# The Association between the Pan-Immune-Inflammation Value and Cancer Prognosis: A Systematic Review and Meta-Analysis

**DOI:** 10.3390/cancers14112675

**Published:** 2022-05-27

**Authors:** Deniz Can Guven, Taha Koray Sahin, Enes Erul, Saadettin Kilickap, Thilo Gambichler, Sercan Aksoy

**Affiliations:** 1Department of Medical Oncology, Hacettepe University Cancer Institute, Ankara 06100, Turkey; saadettin.kilickap@istinye.edu.tr (S.K.); saksoy@hacettepe.edu.tr (S.A.); 2Department of Internal Medicine, Hacettepe University Faculty of Medicine, Ankara 06100, Turkey; koraysahin@hacettepe.edu.tr (T.K.S.); eneserul@hacettepe.edu.tr (E.E.); 3Department of Medical Oncology, Istinye University Faculty of Medicine, Istanbul 34010, Turkey; 4Department of Dermatology, Skin Cancer Center, Ruhr-University Bochum, 44791 Bochum, Germany; t.gambichler@klinikum-bochum.de

**Keywords:** biomarker, cancer, immunotherapy, pan-immune-inflammation value, PIV, targeted therapy

## Abstract

**Simple Summary:**

Growing evidence indicates that blood-count-based compound scores could be used as prognostic biomarkers in cancer as reflectors of uncontrolled inflammation in the tumor microenvironment. Several markers have been developed in this regard, including the recent pan-immune-inflammation value (PIV) that incorporates the levels of blood neutrophil, monocyte, platelet, and lymphocytes. In this paper, we reviewed the association between PIV and overall survival or progression-free survival in cancer from the published studies to date. We observed that higher PIV levels were an adverse prognostic factor consistently across several clinical scenarios, including non-metastatic or metastatic disease and treatment with targeted therapy or immunotherapy. In contrast, the data were limited in patients treated with chemotherapy or patients with non-metastatic disease. The available evidence demonstrates that PIV could be a readily available biomarker for prognosis prediction in cancer. However, further research is needed to explore the promise of PIV as a prognostic biomarker in cancer.

**Abstract:**

Background: Prognostic scores derived from the blood count have garnered significant interest as an indirect measure of the inflammatory pressure in cancer. The recently developed pan-immune-inflammation value (PIV), an equation including the neutrophil, platelet, monocyte, and lymphocyte levels, has been evaluated in several cohorts, although with variations in the tumor types, disease stages, cut-offs, and treatments. Therefore, we evaluated the association between survival and PIV in cancer, performing a systematic review and meta-analysis. Methods: We conducted a systematic review from the Pubmed, Medline, and Embase databases to filter the published studies until 17 May 2022. The meta-analyses were performed with the generic inverse-variance method with a random-effects model. Results: Fifteen studies encompassing 4942 patients were included. In the pooled analysis of fifteen studies, the patients with higher PIV levels had significantly increased risk of death than those with lower PIV levels (HR: 2.00, 95% CI: 1.51–2.64, *p* < 0.001) and increased risk of progression or death (HR: 1.80, 95% CI: 1.39–2.32, *p* < 0.001). Analyses were consistent across several clinical scenarios, including non-metastatic or metastatic disease, different cut-offs (500, 400, and 300), and treatment with targeted therapy or immunotherapy (*p* < 0.001 for each). Conclusion: The available evidence demonstrates that PIV could be a prognostic biomarker in cancer. However, further research is needed to explore the promise of PIV as a prognostic biomarker in patients with non-metastatic disease or patients treated without immunotherapy or targeted therapy.

## 1. Introduction

Cancer is a global health problem and a leading cause of mortality worldwide [1,2]. Around 20 million people are diagnosed with cancer, and 10 million cancer-related deaths are recorded yearly [3]. Additionally, the global burden of cancer morbidity is increasing due to cancer itself and the short- and long-term toxicities of anti-cancer treatments in cancer survivors [4,5]. Although the rate of survival has improved in the last decade due to enhancements in novel treatment options in most cancers, the outcomes are still far from desired. Therefore, there is an urgent need for development of biomarkers for better treatment customization. However, the advancements in biomarker development have significantly lagged behind the developments in the treatment field due to complexities in tissue availability and test platforms, as well as cost issues [6,7,8,9].

Inflammation is vital for innate immunity and is required for immune surveillance and clearance of external insults to prevent them from harming the host [10,11]. The inflammatory response of the innate immune system constitutes the first line of defense against carcinogens; thus, impairments in this response could create a predisposition for cancer development and progression [12,13]. The use of this inflammatory response for cancer treatment was first hypothesized in the 19th century when William Coley found that inducing inflammation with the inoculation of killed bacteria to the areas of sarcoma results in tumor shrinkage in some patients [14]. Later, the discovery of intravesical Bacillus Calmette–Guérin (BCG) for bladder cancer, which generates a local inflammatory immune reaction in the bladder, substantially supported inducing immune activation as an anti-cancer strategy [15]. Additionally, the increased cancer risk in patients with primary [16] and secondary immune deficiencies such as Acquired Immune Deficiency Syndrome (AIDS) [17], and transplant recipients [18,19], further supported the importance of immune surveillance in the promotion and progression of cancer and suggested the potential of immune activation as an anti-cancer treatment strategy [16]. 

Notably, although inflammation is vital for cancer prevention, uncontrolled inflammation could have the opposite effect on cancer development due to several mechanisms, including DNA damage by the pro-inflammatory cytokines and chemokines and the increased risk of genomic alterations and instability [20]. Additionally, chronic antigenic stimulation due to chronic inflammation could lead to uncontrolled T stimulation and autoimmunity. Thus, T-cells express more checkpoint inhibitors such as programmed cell death-ligand 1 (PD-1) and cytotoxic T lymphocyte-associated antigen-4 (CTLA-4), exhibit poor antigen response to avoid autoimmunity, and enter a quiescence state also known as T-cell exhaustion [21]. Due to the reasons cited above (T-cell exhaustion and DNA damage), inflammation is accepted as a hallmark of cancer and is associated with multiple stages of cancer promotion and progression, including angiogenesis, invasion, progression, and metastasis [22]. Furthermore, in epidemiological studies, chronic inflammation secondary to infections and autoimmune disorders is linked to several types of cancer (e.g., Helicobacter pylori with gastric carcinoma, hepatitis B and C with hepatocellular carcinoma, and human papillomavirus with cervical cancer, inflammatory bowel disease with colorectal cancer (CRC)) [20].

In this long process, tumor cells express more PD-L1 and CTLA-4 and reduce the expression of major histocompatibility complex (MHC) class I tumor-associated antigens to become less immunogenic and hence avoid immune identification [23]. In the escape phase of immunoediting, tumor cells release soluble factors, such as the enzyme indoleamine 2,3-dioxygenase (IDO), that promote the activity of immunosuppressive leukocytes, including T regulatory (Treg) cells. Therefore, tumor cells develop the ability to reduce or manipulate immune responses [23]. The knowledge of the interplay between tumor and host immunity, as well as the mechanisms that regulate T-cell activity, has led to the development of cancer immunotherapy. Treatment aims to re-invigorate exhausted tumor-infiltrating T lymphocytes (TILs) to destroy tumor cells by reducing immune regulatory inhibition rather than targeting particular molecules in tumors [24]. Besides exploiting the therapeutic role of inflammation and the immune system in cancer, the measurement of uncontrolled inflammatory pressure could be a biologically plausible biomarker in predicting cancer prognosis and planning anti-cancer treatment. However, the optimal measurement of this inflammatory pressure and its quantification are relatively unknown.

Recently, compound prognostic scores derived from peripheral blood count indices have garnered significant interest as indirect measures of inflammatory pressure in cancer [25,26]. These scores generally involve dividing the counts of pro-inflammatory cells such as neutrophils, platelets, and monocytes to the lymphocytes, the main drivers of anti-cancer immunity in the tumor vicinity [27]. Several studies with neutrophil-to-lymphocyte ratio (NLR) and platelet-to-lymphocyte ratio (PLR) have been conducted, and they have consistently reported poorer survivals with higher NLR or PLRs [28,29,30,31], although these indices (NLR and PLR) only evaluated the counts of two immune-inflammatory cells. This issue led to the development of a novel score, the pan-immune-inflammation value (PIV), an equation that includes the neutrophil, platelet, and monocyte levels in addition to lymphocytes [32]. After the pilot study in CRC published by Fuca et al. in 2020 [32], several studies on other tumors have evaluated the association between PIV levels and cancer prognosis, although the tumor types, disease stages, cut-offs, methodology, and reporting varied considerably. In the present study, we evaluated the association between survival outcomes and PIV levels in human cancers by performing a systematic review and meta-analysis.

## 2. Materials and Methods

### 2.1. Literature Search

We conducted a systematic review following the Preferred Reporting Items for Systematic Reviews and Meta-analysis guidance (PRISMA) [33]. This protocol was registered with the Open Science Framework (OSF) at http://doi.org/10.17605/OSF.IO/A486H (accessed on 21 May 2022). The Web of Science, PubMed, and Embase databases were used to systematically filter the published studies from inception to 17 May 2022 for this systemic review. The selected MeSH search term was “pan-immune-inflammation value” to prevent missing out on published studies with a more specific search strategy. 

### 2.2. Inclusion and Exclusion Criteria

The included studies meet the following criteria: (1) prospective and retrospective study to investigate the prognostic effects of PIV on patients with cancer; (2) the patients were graded strictly according to the definition of PIV, and were grouped clearly; (3) articles containing the hazard ratio (HR) of overall survival (OS), disease-free survival (DFS) or progression-free survival (PFS); and (4) the full text was available in English. Meanwhile, the exclusion criteria were as follows: (1) duplicated articles; (2) chapters of books, case reports, editorial letters, review articles, and opinion papers; (3) animal studies; (4) studies including patients without cancer; and (5) studies without data for HRs and confidence intervals (CIs).

### 2.3. Study Selection and Data Extraction

Our systematic search retrieved a total of 49 records. After removing the duplications (*n* = 30), we screened the abstracts of the remaining 19 records and excluded 5 records due to missing survival data (*n* = 1) and inclusion of patients without cancer (*n* = 4). We evaluated the full texts of the remaining 14 articles and included these studies in the meta-analyses. An additional article was found and included in the study from the citations of the included articles. The PRISMA diagram for article selection is included in the Appendix A. 

Two authors independently extracted the following data from the available studies (DCG, TKS) following the Meta-analysis of Observational Studies in Epidemiology (MOOSE) guidelines [34]: lead author names, year of publication, total number of patients, and adjusted HR with 95% CIs for OS and DFS or PFS. Due to the definition of the same events in different settings, we used the DFS/PFS term for the DFS or PFS events, as previously suggested [35]. The authors of three studies were contacted via mail to provide the HRs that were not available in the publication. Two authors (DCG and TKS) independently reviewed and extracted the available data, and any disagreements were resolved by a discussion with the senior author (SA). The individual study qualities and risk of bias were evaluated independently by two authors (DCG and EE) using the Newcastle-Ottawa Scale.

### 2.4. Meta-Analyses

The primary objective of this study was to evaluate the association between the OS or DFS/PFS and PIV levels in cancer. The secondary objective was to evaluate the association between the OS or DFS/PFS according to disease stage (metastatic or non-metastatic), treatment type (targeted therapy or immunotherapy), and PIV cut-off (300, 400, and 500). We conducted further subgroup analyses according to treatment type due to heterogeneity of the included studies (immunotherapy monotherapy for the immunotherapy subgroup, and oral targeted therapy subgroup). We were unable to conduct additional subgroup analyses in immunotherapy-immunotherapy and chemo-immunotherapy combinations due to the lack of separate data and the presence of only one related study, respectively.

We performed the meta-analyses with the generic inverse-variance method with a random effects model considering the high degree of heterogeneity in the analyses. We used HRs with 95% two-sided CIs as the principal summary measure and reported the heterogeneity within each subgroup with I-square statistics. Moreover, we conducted additional analyses with the fixed effects model after the exclusion of studies that caused a high degree of heterogeneity. We conducted the meta-analyses using the Review Manager software, version 5.4 (The Nordic Cochrane Center, The Cochrane Collaboration, Copenhagen, Denmark) and considered *p* values below as 0.05 statistically significant.

## 3. Results

### 3.1. The Study Characteristics

Fifteen studies encompassing 4942 patients were included in the meta-analyses [32,36,37,38,39,40,41,42,43,44,45,46,47,48,49]. The available studies were conducted on several tumors, including CRC, melanoma, breast cancer, and non-small cell lung cancer (Table 1). Additionally, one study included a basket cohort consisting of several types of tumors treated with immunotherapy [39]. The sample sizes varied between 49 [43] and 1312 [47], and the PIV cut-offs ranged from 285 [41] to 600 [37]. Median levels, receptor operating characteristics (ROC), and maximally selected rank statistics were used in each of the five studies. In two studies using ROC analyses, a significant association was not found between PIV and several outcome measures [42,43]. Most studies were focused on the prognosis in immunotherapy-treated cohorts and metastatic disease (Table 1). One study evaluated both patients treated with immunotherapy (nivolumab and pembrolizumab alone) and targeted therapy with BRAF inhibitors/MEK inhibitors, and separate data for these treatment types were included in the subgroup analyses [32]. Ten studies included Caucasian patients [32,36,37,38,39,40,41,42,43,48], while five studies were conducted in the Far East [44,45,46,47,49]. Most studies had a low risk of bias according to the NOS (Table 2).

### 3.2. Association between OS and PIV Levels

All but two studies reported a negative effect of higher PIV levels on OS [42,43]. In the pooled analysis of 15 studies, the patients with higher PIV levels had a significantly increased risk of death than those with lower PIV levels (HR: 2.00, 95% CI: 1.51–2.64, *p* < 0.001) (Figure 1a). The included studies demonstrated a high degree of heterogeneity (I^2^ = 90%). A significant portion of heterogeneity stemmed from the study by Susok et al. [42]. The heterogeneity decreased to 40% with the exclusion of this study. We conducted a fixed-effect meta-analysis after the exclusion of this study and observed consistent results (HR: 2.01, 95% CI: 1.70–2.38, *p* < 0.001) (Appendix A). Additionally, the sensitivity analyses conducted with the subtraction of individual studies demonstrated consistent results.

Subgroup analyses for the disease stage (non-metastatic (HR: 1.97, 95% CI: 1.58–2.45, *p* < 0.001) and metastatic stage (HR: 2.21, 95% CI: 1.47–3.32, *p* < 0.001)) demonstrated a similar negative association between higher PIV levels and OS (Figure 2a,b). Similarly, subgroup analyses for the treatment type (immunotherapy (HR: 2.06, 95% CI: 1.12–3.79, *p* = 0.020) and targeted therapy (HR: 3.41, 95% CI: 1.96–5.92, *p* < 0.001)) demonstrated consistent results (test for subgroup differences, *p* = 0.230) (Figure 3a,b). Furthermore, the subgroup analysis in patients treated with immunotherapy monotherapy (HR: 2.48, 95% CI: 1.13–5.42, *p* = 0.020) demonstrated a higher risk of death in patients with higher PIV levels, similar to the patients with higher PIV levels treated with oral targeted therapy (HR: 3.64, 95% CI: 2.40–5.51, *p* < 0.001) (Appendix A). The analyses according to variable PIV cut-offs demonstrated consistent results: cut-off 500 (HR: 1.95, 1.45–2.63, *p* < 0.001), cut-off 400 (HR: 2.04, 95% CI: 1.67–2.50, *p* < 0.001), cut-off 300 (HR: 2.04, 95 CI: 1.78–2.33, *p* < 0.001) (Figure 4a,b).

### 3.3. Association between DFS/PFS and PIV Levels

Fourteen studies with available DFS/PFS data were included in the analyses [32,36,37,38,39,40,41,42,43,44,45,46,48,49]. All but two studies reported significantly lower DFS/PFS with higher PIV levels, while the DFS/PFS difference did not reach statistical significance in the studies by Guven et al. and Susok et al. [39,42]. In the pooled analysis, the patients with higher PIV levels had a significantly increased risk of progression or death than those with lower PIV levels (HR: 1.80, 95% CI: 1.39–2.32, *p* < 0.001) (Figure 1b). Similar to the OS analyses, the heterogeneity across studies significantly decreased with the exclusion of the study by Susok et al. (89% to 28%) [42]. After the exclusion of this study, we conducted a fixed-effect meta-analysis for PFS and observed consistent results (HR: 1.77, 95% CI: 1.57–1.98, *p* < 0.001) (Appendix A). The sensitivity analyses conducted with the subtraction of individual studies also demonstrated consistent results.

Subgroup analyses for the disease stage (non-metastatic (HR: 1.94, 95% CI: 1.39–2.70, *p* < 0.001) and metastatic stage (HR: 1.71, 95% CI: 1.27–2.30, *p* < 0.001)) (Figure 2a,b) and treatment type (immunotherapy (HR: 1.63, 95% CI: 1.06–2.53, *p* < 0.001) and targeted therapy (HR: 2.39, 95% CI: 1.64–3.50, *p* < 0.001)) demonstrated consistent results (test for subgroup differences, *p* = 0.19) (Figure 3a,b). Similarly, the patients with higher PIV levels treated with oral targeted therapy had increased risk of progression or death (HR: 2.92, 95% CI: 2.10–4.07, *p* < 0.001). By contrast, the association between PIV levels and DFS/PFS did not reach statistical significance in the pooled analysis of two studies (HR: 1.86, 95% CI: 0.95–3.65, *p* = 0.070) (Appendix A). The analyses results according to the PIV cut-offs were consistent: cut-off 500 (HR: 1.57, 95% CI: 1.20–2.06, *p* = 0.001), cut-off 400 (HR: 1.82, 95% CI: 1.47–2.24, *p* < 0.001), and cut-off 300 (HR: 1.78, 95% CI: 1.54–2.06, *p* < 0.001) (Figure 4a,b).

## 4. Discussion

In this study, we observed a negative association between OS or DFS/PFS and higher PIV levels in a pooled analysis of over 4000 patients. The higher PIV level was a consistent negative prognostic factor consistently across several clinical scenarios, including non-metastatic or metastatic disease and treatment with targeted therapy or immunotherapy. To our best knowledge, the present study is the first meta-analysis evaluating the association between PIV and survival outcomes in cancer.

Uncontrolled inflammation perpetuates the carcinogenesis via modulation of the tumor microenvironment (TME) with the secretion of pro-inflammatory mediators (TNF-alfa, IL-6) and angiogenesis factors, chemotaxis of immune-tolerant tumor-associated macrophages (TAMs), and other stromal auxiliary elements [50,51,52]. Altogether, these factors create an inflammatory pressure in the TME, resulting in immune exhaustion and immune evasion [53,54]. A higher inflammatory pressure evidenced by the increased levels of pro-inflammatory mediators, TAMs, and stromal cells in the TME is consistently correlated with poor outcomes and treatment resistance in pathology studies from several tumors [55,56,57,58].

Although the immune and inflammatory status of the TME could be used as a reflector of tumor behavior and patient prognosis, the need for biopsies and the use of complex platforms are well-known limitations of tissue-based biomarkers [59]. Additionally, the need for tumor- or treatment-specific development and use has reduced the interest in most tissue-based biomarkers. Peripheral blood-based biomarkers have emerged to resolve these limitations, and simple parameters from a complete blood count (CBC) could be used as a biomarkers reflecting TME and tumor behavior [60,61]. The minimally invasive retrieval and the low cost involved make CBC-based biomarkers highly attractive, and a significant body of evidence has developed in the last decade with these biomarkers, particularly with NLR and PLR [31,62,63].

Cancer cells and platelets have important interactions in TME and circulation [64]. It has been found that platelets may play a key role in tumor growth and metastasis via different pathways [65]. Platelets form a thrombus with circulating tumor cells, enabling tumor cells to evade immune system action. Furthermore, activated platelets could secrete a variety of growth factors that aid tumor invasion and development [65]. Similar to platelets, monocytes could be associated with cancer prognosis. Furthermore, blood monocyte counts could reflect tumor-associated macrophages (TAM) in the TME, which are among the main drivers of immunosuppression in the TME [66]. Specifically, M2-type macrophages derived from monocytes affect angiogenesis, invasion, and immunosuppression via the vascular endothelial growth factor (VEGF), tumor necrosis factor-alpha (TNF-α), and interleukin (IL)-10, respectively [67]. Neutrophils have also been linked to tumor growth through the generation of reactive oxygen species and the secretion of pro-tumor chemokines [68,69] Meanwhile, lymphocytes are the primary drivers of anti-cancer immunity in the TME [70]. The PIV score has been created to incorporate several mediators in the immune system to model and reflect the inflammatory pressure more precisely and to prevent fragmented information regarding systemic inflammation [32]. Since all pro-inflammatory cells in the blood count are included in the calculation, PIV has a strong biologic rationale as a biomarker and might potentially result in better risk stratification than NLR or PLR.

The PIV score was recently developed based on the dataset of two phase-III clinical trials and aimed to add on the previously used CBC-based biomarkers [32]. In the pilot study, PIV remained a statistically significant prognostic parameter for PFS and OS in a model that included two other CBC-based indices (NLR and systemic immune-inflammation index). Similarly, in the study by Sahin et al. in localized breast cancer, PIV outperformed NLR, PLR, monocyte-to-lymphocyte ratio (MLR), and the systemic immune-inflammation index, thus underscoring the value of adding more parameters to the prognostic score to reflect inflammatory pressure from CBC [36]. In the study of Fuca et al. in patients with colorectal cancer (CRC), PIV score outperformed other immune-inflammatory biomarkers in the logistic regression [32]. In addition, the PIV score could be incorporated into the compound prognostic scores similar to Gustave Roussy and Royal Marsden scores, as suggested by Guven et al. [39,71,72]. The authors assessed the PILE-composite score of lactate dehydrogenase, Eastern Cooperative Oncology Group performance status (ECOG), and PIV in patients receiving immune checkpoint inhibitors (ICIs). They found that high PILE scores were a risk factor for decreased OS and PFS and may be used as a biomarker for ICIs [39]. Additionally, Corti et al. have evaluated the PIV score change dynamically, and early PIV increase from the baseline was associated with poor ICIs response and survival outcomes in MSI-high CRC patients [38]. Their findings supported the use of PIV changes as a dynamic biomarker. However, Perez-Martelo observed no significant association between early PIV changes and OS or PFS in the metastatic CRC patients treated with first-line chemotherapy, although a statistically significant increase was observed in the PIV levels of patients with eminent progression [48]. From a biological standpoint, the relationship between high PIV and initial resistance to PD-1 blockade and worse prognosis is not surprising considering the origin of myeloid-derived suppressor cells (MDSCs), immune modulatory cell populations associated with resistance to PD-1/PD-L1 inhibition, from monocytes and neutrophils [73]. It is suggested that identifying potentially resistant groups with higher PIV scores may help clinicians implement earlier ICI combinations and treatment intensification to elicit better responses [38]. Furthermore, the work by Perez-Martelo in CRC implies that PIV monitoring might be useful in predicting disease progression earlier [48]. Whether this earlier progression could lead to changes in clinical practice or improve outcomes should be further investigated.

In a study with HER2 (+) advanced breast cancer patients, Ligorio et al. compared other prognostic indices, namely PLR, MLR, and NLR, with the PIV score. A trend toward an association with worse PFS was only observed with the PIV score as the prognostic index in multivariate analyses [41]. Recently, Lin et al. evaluated the efficacy of PIV with the typical TNM staging method in predicting prognosis in patients with breast cancer and found that PIV was more accurate in predicting OS than the traditional TNM staging system, thus emphasizing PIV’s clinical utility [47] As a result, it is considered that the PIV formula, which uses the counts of four types of blood cells (i.e., monocytes, neutrophils, platelets, and lymphocytes), might provide a more consistent and accurate prediction of poor prognosis than the previously recommended indices. Further research with the models incorporating the other CBC-based indexes with PIV is eagerly awaited to delineate the possible superiority of PIV over other CBC-based biomarkers.

The present study is subject to several limitations inherent to the study design and the included studies. First, we included the reported HRs from the studies instead of individual patient data. Additionally, the included studies were heterogeneous in terms of the included tumor types, disease stages, and treatment types, although the subgroup analyses demonstrated a consistent trend across these clinically relevant subgroups. Lastly, several studies included molecularly selected subsets encompassing a small percentage of patients in the relevant tumor type (MSI-H and ALK+), which could limit the generalizability of the results. However, despite these limitations, we provide the first meta-analysis on PIV, a promising and minimally invasive prognostic biomarker in cancer.

## 5. Conclusions

In conclusion, the available evidence demonstrates that PIV could be a minimally invasive prognostic biomarker in several cancers. However, further research is needed to explore the promise of PIV as a prognostic biomarker in patients with non-metastatic disease or patients treated without immunotherapy or targeted therapy.

## Figures and Tables

**Figure 1 cancers-14-02675-f001:**
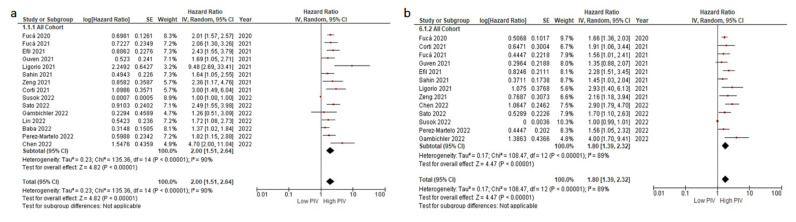
The association between PIV levels and OS (**a**) and DFS/PFS (**b**). Lines (○) indicate 95% CIs. Diamond (♦) indicates the pooled effect size.

**Figure 2 cancers-14-02675-f002:**
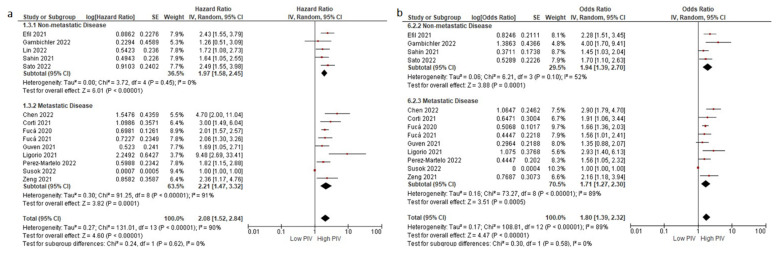
Subgroup analyses according to disease stage in OS (**a**) and DFS/PFS (**b**). Lines (○) indicate 95% CIs. Diamond (♦) indicates the pooled effect size.

**Figure 3 cancers-14-02675-f003:**
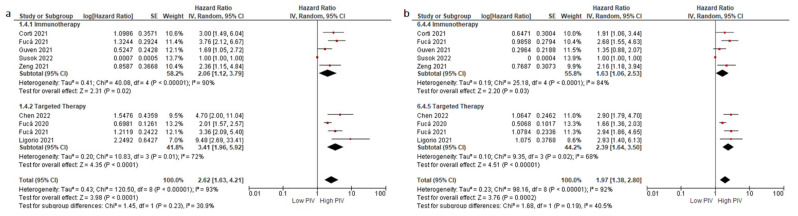
Subgroup analyses according to treatment type in OS (**a**) and DFS/PFS (**b**). Lines (○) indicate 95% CIs. Diamond (♦) indicates the pooled effect size.

**Figure 4 cancers-14-02675-f004:**
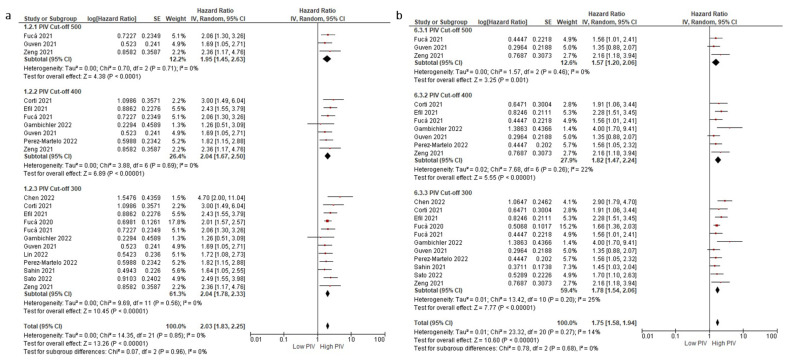
Subgroup analyses according to PIV cut-off in OS (**a**) and DFS/PFS (**b**). Lines (○) indicate 95% CIs. Diamond (♦) indicates the pooled effect size.

**Table 1 cancers-14-02675-t001:** Baseline characteristics of included studies.

Lead Author, Year	Country	Sample Size	Treatment	PIV Cut-Off Value	Cut-Off Selection	Tumor Type	Tumor Stage	Adjustment Factors	Outcome	Additional Comments
Fucà, 2020, [32]	Italy	438	Tribe study: FOLFIRI + Bevacizumab vs. FOLFOXIRI + Bevacizumab Valentino study: mFOLFOX6 + Panitumumab	390	MSR	CRC	IV	- ECOG - Prior adjuvant treatment - Primary tumor resected - Synchronous metastases - Number of metastatic sites - Primary tumor sidedness - RAS/BRAF status - NLR - PLT - Monocyte - SII	- PFS - OS	PIV outperformed the other immune-inflammatory biomarkers in regression model
Corti, 2021, [38]	Italy	163	- Nivolumab plus ipilimumab (32.5%) - Nivolumab (47.3%) - Pembrolizumab (9.8%) - Dostarlimab (10.4%)	492	MSR	CRC	IV	- ECOG - ICI regimen	- PFS - OS - CBR	Early PIV increase was independently correlated with clinical benefit (aOR: 0.23, 95% CI 0.08–0.66, *p* = 0.007)
Fucà, 2021, [37]	Italy	228	ICI: - Nivolumab (61.3%) - Pembrolizumab (38.7%) TT: - Vemurafenib (22.0%) - Dabrafenib (2.8%) - Vemurafenib plus cobimetinib (10.1%) - Dabrafenib plus trametinib (65.1%)	600	MSR	Melanoma	IV	- ECOG - M substage - Metastatic sites - LDH - Steroids use	- PFS - OS - Best response	High PIV was associated with primary resistance to both targeted therapy (OR: 8.42; 95% CI 2.50–34.5, *p* < 0.001) and ICI (OR: 3.98, 95% CI 1.45–12.32, *p* = 0.005)
Guven, 2021, [39]	Turkey	120	- Nivolumab (78.3%) - Atezolizumab (17.5%) - Pembrolizumab (4.2%)	513.4	Median value	RCC, NSCLC, Melanoma, Other	IV	- ECOG - LDH levels - Liver metastasis - BMI category	- PFS - OS	A model combining PIV, ECOG status, and LDH levels (PILE Score) was able to predict 12-week PFS and 24-week OS
Ligorio, 2021, [41]	Italy	57	- Taxane-Transtuzumab - Pertuzumab	285	Median value	Breast Cancer	IV	- Number of metastatic sites - Visceral metastasis - Brain metastasis	- PFS - OS - Response	PIV outperformed MLR, PLR, and NLR in predicting OS
Sahin, 2021, [36]	Turkey	743	- Anthracycline plus taxane (68.6%) - Anthracycline-based regimens (27.5%) - Taxane-based regimens (3.9%)	306.4	ROC curve	Breast Cancer	I-IV	- Clinical T stage - NLR - MLR - PLR - ER status - Her-2 status - Ki-67 index	- pCR - DFS - OS	Pre-treatment PIV appears to be a predictor for pCR and survival, outperforming NLR, MLR, PLR in predicting pCR
Zeng, 2021, [49]	China	53	Control group of NCT03041311 (53 patients): carboplatin, etoposide, and atezolizumab Validation group (84 patients): - PD-1 antibody (29.8%) - PD-L1 antibody (70.2%)	581.95	Median value	SCLC	Extensive Stage	- LDH	- PFS - OS - DCR - DCB	Higher PILE score was associated with worse treatment efficacy (DCR: 84.21% vs. 100%, *p* = 0.047, DCB rate: 10% vs. 48.5%, *p* = 0.060)
Efil, 2021, [40]	Turkey	304	Adjuvant chemotherapy (52%)	491	Median	CRC	II-III	- Age - Stage	- DFS	A model combining PIV and CD8 + TIL density was able to predict DFS
Sato, 2022, [44]	Japan	758	Adjuvant chemotherapy (30%)	376	ROC curve	CRC	I-III	- Age - CA19-9 - CEA - AGR - Post-operative complication	- RFS - OS	A high preoperative PIV was significantly associated with depth of tumor invasion and advanced TNM stage (II, III)
Gambichler, 2022, [43]	Germany	49	N/A	372	ROC curve	MCC	I-III	- Age > 75 - Disease stage - Elevated CRP	- Recurrence - OS	An association between PIV levels and stage was present
Susok, 2022, [42]	Germany	62	- Nivolumab (38.7%) - Pembrolizumab (24.5%) - Ipilimumab (14.5%) - Nivolumab plus Ipilimumab (22.6%)	455	ROC curve	Melanoma	III-IV	N/A	- PFS - DSS - Best response	SII and PIV were not significantly associated with best response to ICI treatment (*p* = 0.87/0.64), PFS (*p* = 0.73/0.91), and melanoma-specific survival (*p* = 0.13/0.17).
Chen, 2022, [45]	China	94	- Crizotinib (89.4%) - Alectinib (10.6%) - Ceritinib (1.0%)	364	Median	Lung Cancer	III-IV	- Liver metastasis	- PFS - OS	Although PIV, NLR, PLR, and SII were associated with poor median OS, only higher PIV was independently associated with poor survival outcomes (HR = 4.70, 95% Cl: 2.00–11.02, *p* < 0.001).
Baba, 2022, [46]	Japan	433 (Validation Cohort)	N/A	164.6	ROC	Esophageal Cancer	I-IV	- Preoperative therapy - Pathological stage	- OS	The PIV-high cases were significantly associated with a low TIL status (*p* < 0.001) and low CD8-positive cell counts (*p* = 0.011)
Lin, 2022, [47]	China	1312	Adjuvant chemotherapy (81.3%)	310.2	MSR	Breast Cancer	I-III	- Stage (T and N) - PR status - Ki-67 - Histopathological type	- OS	The prognostic model showed a good discriminating ability for OS prediction, with a C-index of 0.759 (95% CI 0.715–0.802)
Perez-Martelo, 2022, [48]	Spain	130	- Oxaliplatin-based regimen (74%) - Non-oxaliplatin-based regimen (26%)	424.05	MSR	CRC	IV	- CEA - ECOG-PS - Primary tumor location - Lymph node metastases - Primary tumor resection	- PFS - OS - DCR - ORR	- Baseline PIV was not correlated either with DCR or ORR

Abbreviations: MSR: maximally selected rank statistics; ECOG: Eastern Cooperative Oncology Group; mCRC: metastatic colorectal cancer; NLR: neutrophil-to-lymphocyte ratio; PLR: platelet-to-lymphocyte ratio; PLT: platelet count; MONO: monocyte count; SII: systemic immune-inflammation index; PIV: Pan-Immune-Inflammation Value; OR: odds ratio; CBR: Clinical Benefit Rate; DCR: Disease Control Rate; DCB: durable clinical benefit; DSS: disease specific survival; TIL: tumor-infiltrating lymphocytes; ROC: Receiver Operating Characteristic; MCC: Merkel cell carcinoma; BMI: body mass index; ICI: immune checkpoint inhibitor; HR: hazard ratio; CBC: complete blood count; CRP: C-reactive protein; ER: estrogen receptor; CEA: carcinoembriyonic antigen; TT: Targeted Therapy; PR: progesterone receptor.

**Table 2 cancers-14-02675-t002:** Newcastle-Ottawa scores of included studies (Note: A star system was used for allow a semi quantitative assessment of study quality. A study was awarded a maximum of four stars for the selection and three stars for exposure/outcome categories. A maximum of two stars were awarded for comparability).

Lead Author, Year	Selection	Comparability	Exposure/Outcome	Reference
Fucà, 2020	****	**	**	[32]
Corti, 2021	***	**	***	[38]
Fucà, 2021	****	**	***	[37]
Guven, 2021	****	**	***	[39]
Ligorio, 2021	***	**	***	[41]
Sahin, 2021	***	**	***	[36]
Zeng, 2021	****	**	***	[49]
Efil, 2021	No full-text data available			[40]
Sato, 2022	***	**	***	[44]
Gambichler, 2022	***	**	***	[43]
Susok, 2022	***	*	**	[42]
Chen, 2022	****	**	**	[45]
Baba, 2022	****	**	**	[46]
Lin, 2022	****	**	***	[47]
Perez-Martelo, 2022	****	**	***	[48]

## Data Availability

All relevant data are included in the study and Appendix A.

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
