# Peer review of "The Association between the Pan-Immune-Inflammation Value and Cancer Prognosis: A Systematic Review and Meta-Analysis"

_cancers, 2022, doi:10.3390/cancers14112675_

Round 1

Reviewer 1 Report

The paper by  Can Guven et al. provides a comprehensive overview on the prognostic role of PIV in solid tumors; data are convincing and findings are of interest. However, I have some concerns over the following points:

  1. Authors state that HRs for OS and DFS were extracted from each study, and results are expressed as pooled HRs after performing a meta analysis. However, according to figures, odds ratio were used to conduct the analysis. Figures should be corrected. 
  2. In the abstract and results authors state that patients with higher PIV have lower OS/DFS. The correct way to state these findings would be that patients with higher PIV have an increased risk of death or progression: meta analysis gives an estime of the risk; it does not provide quantitative information about survival. 
  3. The introduction is too generic and poorly written. The role of chronic inflammation in inducing immune exhaustion and resistance to anti-cancer treatment should be clearly explained, also mentioning the importance of inflammation in the "immunotherapy era". I would suggest to rewrite the introduction following a more logical structure (e.g general role of inflammation in preventing cancer mentioning the association between immune suppression and cancer; how, on the other side, chronic inflammation induces cancer; how cancer evades immune surveillance; how it can be treated).
  4. In the methods, authors should clearly state the inclusion criteria of the meta-analysis. I would advise to write  them point-by-point, e.g.: we selected the studies meeting the following inclusion criteria 1...2...3. Please define better what does "relevant survival data" stand for.  Please include the exact timeframe (from... to March 2022) the research was restricted to. Was the meta analysis submitted on PROSPERO? If not, why? If yes, please provide the reference number.
  5. In the results I would specify in details which treatment patients received in each study. This can be added to table 1, I would avoid generic terms like target therapy (which one? anti BRAF and anti MEK?) or ICI (single or double? Alone or with chemotherapy?).
  6.  In the results, authors correctly performed a subgroup analysis based on tumor stage and treatment type. However, the attribution to each treatment subgroup is not clear! In the study by Fuca et al. patients received a chemotherapy backbone, either in association with panitumumab (Valentino) or bevacizumab (TRIBE): it might be misleading to include these patients along with those receiving targeted therapy for melanoma. Similarly, in the study by Zeng et al,  SCLC patients received immunotherapy in combination with chemotherapy. The authors should acknowledge these differences and clearly explained the criteria they used to define each subgroup. 
  7. What do "localized and advanced stages" stand for? I assume that advanced stage refers to metastatic status. What about localized? Does it include surgically resected patient ? Does it include patients receiving adjuvant/neoadjuvant treatments? I would suggest to clearly report what do authors mean for localized and advanced stages.
  8. The discussion is not clear. I think that several important points are missing, and should be discussed. Why inflammation is associated with worse prognosis in several types of cancer? Why should PIV perform better than other index? Is there a biological rationale beyond this?  How could these markers be adopted in clinical practice? Is there any information about their predictive role? What does this paper add to already existing data? Authors conclude that this is the first meta analysis on this topic. Why did they decide to do a meta analysis, why is this important?

Author Response

May 25, 2022

Dear Editor,

Thank you very much for your kind letter of May 11, 2022 regarding our manuscript cancers-16940625, entitled "The Association between the Pan-Immune-Inflammation Value and Cancer Prognosis: A Systematic Review and Meta-Analysis".

We thank the reviewers for the constructive criticism and insightful comments. Thank you very much for your comments. This manuscript was edited for proper English language, grammar, punctuation, spelling, and overall style by one or more of the highly qualified native English speaking editors. We diligently worked to constructively address each of these comments. This manuscript has been read and approved by all the authors. Our responses to the reviewer’s comments are given point by point:

Comments from the Editors and Reviewers:

General Comments of Reviewer 1.  “The paper by  Can Guven et al. provides a comprehensive overview on the prognostic role of PIV in solid tumors; data are convincing and findings are of interest. However, I have some concerns over the following points:.” 

 Response:  We thank the reviewer for the overall very favorable review of our manuscript and the constructive comments.  The incorporation of the comments and corresponding revisions have enhanced the quality of our manuscript.

Comment #1 of Reviewer 1. Authors state that HRs for OS and DFS were extracted from each study, and results are expressed as pooled HRs after performing a meta analysis. However, according to figures, odds ratio were used to conduct the analysis. Figures should be corrected.”

Response:  We really appreciated the attention of the reviewer and contribution to the article. We corrected that mistake in the Figures.

Comment #2 of Reviewer 1. In the abstract and results authors state that patients with higher PIV have lower OS/DFS. The correct way to state these findings would be that patients with higher PIV have an increased risk of death or progression: meta analysis gives an estime of the risk; it does not provide quantitative information about survival.

Response:  The reviewer’s point is well taken. These findings have been revised as “ In the pooled analysis of fifteen studies, the patients with higher PIV levels had significantly increased risk of death compared to lower PIV levels (HR: 2.00, 95% CI: 1.51-2.64, p<0.001) (Figure 1a). In the pooled analysis, the patients with higher PIV levels had significantly increased risk of progression or death compared to lower PIV levels (HR: 1.80, 95% CI: 1.39-2.32, p<0.001) (Figure-1b).” in the abstract and results.

Comment #3 of Reviewer 1. “The introduction is too generic and poorly written. The role of chronic inflammation in inducing immune exhaustion and resistance to anti-cancer treatment should be clearly explained, also mentioning the importance of inflammation in the "immunotherapy era". I would suggest to rewrite the introduction following a more logical structure (e.g general role of inflammation in preventing cancer mentioning the association between immune suppression and cancer; how, on the other side, chronic inflammation induces cancer; how cancer evades immune surveillance; how it can be treated).”

Response:  Thank you very much for this constructive comment. In order to constructively address Comment #2 of the reviewer, we added the following information to the introduction section:

The inflammation is vital for innate immunity and is required for immune surveillance and clearance of external insults to prevent them from harming the host [10, 11]. The inflammatory response of the innate immune system constitutes the first line of defense against carcinogens; thus, impairments in this response could create a predisposition for cancer development and progression [12, 13]. The use of this inflammatory response for cancer treatment was first hypothesized in the 19th century when William Coley found that inducing inflammation with the inoculation of killed bacteria to the areas of sarcoma results in tumor shrinkage in some patients [14]. Later, the discovery of intravesical Bacillus Calmette–Guérin (BCG) for bladder cancer, which generates a local inflammatory immune reaction in the bladder, substantially supported inducing immune activation as an anti-cancer strategy [15]. Additionally, the increased cancer risk in patients with primary [16] and secondary immune deficiencies such as Acquired Immune Deficiency Syndrome (AIDS)[17], and transplant recipients [18, 19], further supported the importance of immune surveillance in the promotion and progression of cancer and suggested the potential of immune activation as an anti-cancer treatment strategy [20].

Notably, although the inflammation is vital for cancer prevention, the uncontrolled inflammation could have the opposite effect on cancer development due to several mechanisms, including the DNA damage by the pro-inflammatory cytokines and chemokines and the increased risk of genomic alterations and instability [21]. Additionally, chronic antigenic stimulation due to chronic inflammation could lead to uncontrolled T stimulation and autoimmunity. Thus, T-cells express more checkpoint inhibitors like programmed cell death-ligand 1 (PD-1) and cytotoxic T lymphocyte-associated antigen-4 (CTLA-4), exhibit poor antigen response to avoid autoimmunity, and enter a quiescence state also known as T-cell exhaustion [22]. Due to the reasons cited above (T-cell exhaustion and DNA damage), inflammation is accepted as a hallmark of cancer and is associated with multiple stages of cancer promotion and progression, including angiogenesis, invasion, progression, and metastasis [23]. Furthermore, in epidemiological studies, chronic inflammation secondary to infections and autoimmune disorders is linked to several types of cancer (e.g., Helicobacter pylori with gastric carcinoma, hepatitis B and C with hepatocellular carcinoma, and human papillomavirus with cervical cancer, inflammatory bowel disease with colorectal cancer(CRC)) [21]

In this long process, tumor cells express more PD-L1 and CTLA-4 and reduce the expression of major histocompatibility complex (MHC) class I tumor-associated antigens to become less immunogenic and hence avoid immune identification [24]. In the escape phase of immunoediting, tumor cells release soluble factors, such as the enzyme indoleamine 2,3-dioxygenase (IDO), that promote the activity of immunosuppressive leukocytes, including T regulatory (Treg) cells. Therefore, tumor cells develop the ability to reduce or manipulate immune responses [24]. The knowledge of the interplay between tumor and host immunity, as well as the mechanisms that regulate T-cell activity, has led to the development of cancer immunotherapy. Treatment aims to re-invigorate exhausted tumor-infiltrating T lymphocytes (TILs) to destroy tumor cells by reducing immune regulatory inhibition rather than targeting particular molecules in tumors [25]. Besides exploiting the therapeutic role of inflammation and the immune system in cancer, the measurement of uncontrolled inflammatory pressure could be a biologically plausible biomarker in predicting cancer prognosis and planning anti-cancer treatment. However, the optimal measurement of this inflammatory pressure and its quantification are relatively unknown.

Comment #4 of Reviewer 1. “In the methods, authors should clearly state the inclusion criteria of the meta-analysis. I would advise to write  them point-by-point, e.g.: we selected the studies meeting the following inclusion criteria 1...2...3. Please define better what does "relevant survival data" stand for.  Please include the exact timeframe (from... to March 2022) the research was restricted to. Was the meta analysis submitted on PROSPERO? If not, why? If yes, please provide the reference number. “

Response: Thank you very much for the suggestion.  In order to constructively address Comment #4 of the reviewer, we added the following information to the methods section:  2.2 Inclusion and Exclusion Criteria:  The studies which were included must meet the following criteria: 1) prospective and retrospective study to investigate the prognostic effects of PIV on patients with solid cancer; 2) the patients were graded strictly according to the definition of PIV, and the patients were grouped clearly; 3) articles containing both the hazard ratio (HR) of overall survival (OS) or disease-free (DFS) or progression-free survival (PFS); 4) the full text was available in English.  Furthermore, the exclusion criteria were as follows: (1) duplicated articles; (2) chapters of books, case reports, editorial letters, review articles, and opinion papers; 3) studies of animals 4) studies including patients without cancer 5) studies without enough data for HRs and CIs.“. The timeframe has been updated as “ from inception to 17 May 2022…“. We included 2 more recent studies that met the inclusion and exclusion criteria in the meta-analysis (DOI: 10.3389/fonc.2022.830138 and DOI: 10.1038/s41598-022-10884-8 ). This protocol was registered with Open Science Framework (OSF) at http://doi.org/10.17605/OSF.IO/A486H (accessed date: 25 May 2022). We provided this information in the methods section.

Comment #5 of Reviewer 1. “ In the results I would specify in details which treatment patients received in each study. This can be added to table 1, I would avoid generic terms like target therapy (which one? anti BRAF and anti MEK?) or ICI (single or double? Alone or with chemotherapy?). “

Response: Thank you very much for this constructive comment.  We have included  treatment details in Table-1 and tried to avoid generic terms. We could not be able to conduct additional subgroup analyses in immunotherapy-immunotherapy or chemo-immunotherapy combinations due to lack of separate data and presence of only one study, respectively.

Comment #6 of Reviewer 1. “ In the results, authors correctly performed a subgroup analysis based on tumor stage and treatment type. However, the attribution to each treatment subgroup is not clear! In the study by Fuca et al. patients received a chemotherapy backbone, either in association with panitumumab (Valentino) or bevacizumab (TRIBE): it might be misleading to include these patients along with those receiving targeted therapy for melanoma. Similarly, in the study by Zeng et al,  SCLC patients received immunotherapy in combination with chemotherapy. The authors should acknowledge these differences and clearly explained the criteria they used to define each subgroup. “

Response: Thank you very much for the suggestion.  In order to constructively address Comment #6 of the reviewer, we added the following information to the methods section and performed further subgroup analyses according to treatment type.  

“We conducted further subgroup analyses according to treatment type due to heterogeneity of the included studies (immunotherapy monotherapy for immunotherapy subgroup, and oral targeted therapy subgroup).”

Comment #7 of Reviewer 1. “ What do "localized and advanced stages" stand for? I assume that advanced stage refers to metastatic status. What about localized? Does it include surgically resected patient ? Does it include patients receiving adjuvant/neoadjuvant treatments? I would suggest to clearly report what do authors mean for localized and advanced stages. “

Response: The reviewer’s point is well taken. To remove this ambiguity, the “localized “ has been revised as “nonmetastatic“ and the “advanced “ has been revised as “metastatic“ both in the  text and figure.

Comment #8 of Reviewer 1. “The discussion is not clear. I think that several important points are missing, and should be discussed. Why inflammation is associated with worse prognosis in several types of cancer? Why should PIV perform better than other index? Is there a biological rationale beyond this?  How could these markers be adopted in clinical practice? Is there any information about their predictive role? What does this paper add to already existing data? Authors conclude that this is the first meta analysis on this topic. Why did they decide to do a meta analysis, why is this important?”

Response:We really appreciated the contribution of the reviewer to the article. We added the following information to the revised “Discussion” section for further clarification: Although the immune and inflammatory status of the TME could be used as a reflector of tumor behavior and patient prognosis, the need for biopsies and the use of complex platforms are well-known limitations of tissue-based biomarkers [60]. Additionally, the need for a tumor or treatment-specific development and use has reduced the interest in most tissue-based biomarkers. Peripheral blood-based biomarkers have emerged to resolve these limitations, and the simple parameters from the complete blood count (CBC) could be used as a biomarkers reflecting TME and tumor behavior [61, 62]. The minimally invasive retrieval and the low cost involved make CBC-based biomarkers highly attractive, and a significant body of evidence has developed in the last decade with these biomarkers, particularly with NLR and PLR [33, 63, 64].

Cancer cells and platelets have important interactions in TME, and circulation [65]. It is found that platelets may play a key role in tumor growth and metastasis via different pathways [66]. Platelets form thrombus with circulating tumor cells, which enabling tumor cells to evade immune system action. Furthermore, activated plateletscould secrete a variety of growth factors that aid tumor invasion and development [66]. Similar to platelets, monocytes could beassociated with cancer prognosis. Furthermore,blood monocyte counts could reflect tumor-associated macrophages (TAM) in the TME, which are among the main drivers of immunosuppression in TME [67]. Specifically, M2-type macrophages derived from monocytes affect the angiogenesis, invasion, and immunosuppression via the vascular endothelial growth factor (VEGF), tumor necrosis factor-alpha (TNF-α), and interleukin (IL)-10 respectively [68]. Neutrophils have also been linked to tumor growth through the generation of reactive oxygen species and the secretion of pro-tumor chemokines [69, 70] Meanwhile, lymphocytes are the primary drivers of the anti-cancer immunity in the TME [71]. The PIV score has been created to incorporate several mediators in the immune system to the model to reflect the inflammatory pressure more precisely and to prevent fragmented information regarding systemic inflammation[34]. Since all pro-inflammatory cells in the blood count are included in the calculation, the PIV has a strong biologic rationale as a biomarker and might potentially result in better risk stratification than NLR or PLR.

The PIV score was recently developed based on the dataset of two phase-III clinical trials and aimed to add on the previously used CBC-based biomarkers [34]. In the pilot study, the PIV remained a statistically significant prognostic parameter for PFS and OS in a model that included two other CBC-based indices (NLR and systemic immune-inflammation index). Similarly, in the study by Sahin et al. in localized breast cancer, PIV outperformed NLR, PLR, monocyte-to-lymphocyte ratio (MLR), and the systemic immune-inflammation index, thus underscoring the value of adding more parameters to the prognostic score to reflect inflammatory pressure from CBC [37]. In the study of Fuca et al. in patients with colorectal cancer (CRC), PIV score outperformed other immune-inflammatory biomarkers in the logistic regression [34]. In addition the PIV score could be incorporated into the compound prognostic scores similar to Gustave-Roussy and Royal Marsden scores as suggested by Guven et al [40, 72, 73]. The authors assessed the PILE-composite score of lactate dehydrogenase, Eastern Cooperative Oncology Group performance status (ECOG), and PIV in patients receiving immune checkpoint inhibitors (ICIs). They have found that high PILE scores were a risk factor for decreased OS and PFS and may be used as a biomarker for ICIs [40]. Additionally, Corti et al. have evaluated the PIV score change dynamically, and early PIV increase from the baseline was associated with poor ICIs response and survival outcomes in MSI-high CRC patients [39]. Their findings supported the use of PIV changes as a dynamic biomarker. However, Perez-Martelo observed no significant association between early PIV changes and OS or PFS in the metastatic CRC patients treated with first-line chemotherapy, although a statistically significant increase was observed in the PIV levels of patients with eminent progression [49]. From a biological standpoint, the relationship between high PIV and initial resistance to PD-1 blockade and worse prognosis is not surprising considering the origin of myeloid-derived suppressor cells (MDSCs),  immune modulatory cell populations associated with resistance to PD-1/PD-L1 inhibition, from monocytes and neutrophils [74]. It is suggested that identifying potentially resistant groups with higher PIV scores may help clinicians implement earlier ICIs combinations and treatment intensification to elicit better responses [39]. Furthermore,the work by Perez-Martelo in CRC implies that PIV monitoring might be useful in predicting disease progression earlier [49]. Whether this earlier progression could lead to changes in clinical practice or improve outcomes should be further investigated.

In a study with HER2 (+) advanced breast cancer patients, Ligorio et al. have compared other prognostic indices, namely, PLR, MLR, and NLR, with the PIV score.  A trend toward an association with worse PFS was only observed with the PIV score as the prognostic index in multivariate analyses [42]. Recently, Lin et al. have evaluated the efficacy of PIV with the typical TNM staging method in predicting prognosis in patients with breast cancer and found that PIV was more accurate in predicting OS than the traditional TNM staging system, thus emphasizing PIV's clinical utility [48] As a result, it is considered that the PIV formula, which uses the counts of four types of blood cells (i.e., monocytes, neutrophils, platelets, and lymphocytes), might provide a more consistent and accurate prediction of poor prognosis than the previously recommended indices. Further research with the models incorporating the other CBC-based indexes with PIV is eagerly awaited to delineate the possible superiority of PIV over other CBC-based biomarkers. ”

We believe our paper adds important new information and perspective to the field. We believe that all review points have been addressed and hope that the revisions are satisfactory.  We kindly request that our revised manuscript be considered for publication in Cancers.

Thank you very much for your kind consideration.

Kind regards,

Deniz Can Guven

Hacettepe University Cancer Institute, Department of Medical Oncology

06100 Sıhhıye, Ankara, TURKEY

Tel: +90 312 305 43 30

Fax: +90 312 310 01 95

E-mail: denizcguven@hotmail.com

Reviewer 2 Report

The Authors of cancers-1694062 prepared the systematic review of research papers concernig pan-immune inflammation value (PIV) and they used these data for meta-analysis to assess the usefulness of PIV in cancer prognosis. The methodology applied by them is correct and it complies the PRISMA guideline. Applied statistical methods are also correct which makes the results reliable. Their conslusions are justified and emphasize the main advantage of PIV (small invasiveness) as well as its limitation (not enough data to claim PIV as really informative prognostic biomarker). In my opinion this paper can be interesting for general audience. 

Author Response

May 25, 2022

Dear Editor,

Thank you very much for your kind letter of May 11, 2022 regarding our manuscript cancers-16940625, entitled "The Association between the Pan-Immune-Inflammation Value and Cancer Prognosis: A Systematic Review and Meta-Analysis".

We thank the reviewers for the constructive criticism and insightful comments. Thank you very much for your comments. This manuscript was edited for proper English language, grammar, punctuation, spelling, and overall style by one or more of the highly qualified native English speaking editors. We diligently worked to constructively address each of these comments. This manuscript has been read and approved by all the authors. Our responses to the reviewer’s comments are given point by point:

Comments from the Editors and Reviewers:

General Comments of Reviewer 2.  “The Authors of cancers-1694062 prepared the systematic review of research papers concerning pan-immune inflammation value (PIV) and they used these data for meta-analysis to assess the usefulness of PIV in cancer prognosis. The methodology applied by them is correct and it complies the PRISMA guideline. Applied statistical methods are also correct which makes the results reliable. Their conclusions are justified and emphasize the main advantage of PIV (small invasiveness) as well as its limitation (not enough data to claim PIV as a really informative prognostic biomarker). In my opinion this paper can be interesting for general audience.” 

Response: We would like to thank the reviewer for his/her positive and very thorough comments on the manuscript. We are pleased to hear that the manuscript is found to be highly interesting and important in the subject matter.

We believe our paper adds important new information and perspective to the field. We believe that all review points have been addressed and hope that the revisions are satisfactory.  We kindly request that our revised manuscript be considered for publication in Cancers.

Thank you very much for your kind consideration.

Kind regards,

Deniz Can Guven

Hacettepe University Cancer Institute, Department of Medical Oncology

06100 Sıhhıye, Ankara, TURKEY

Tel: +90 312 305 43 30

Fax: +90 312 310 01 95

E-mail: denizcguven@hotmail.com

Round 2

Reviewer 1 Report

The current version of the paper is significantly improved. Authors have adequately addressed all the comments.

I believe it can be accepted in the current form.